# Reduction of Specular Reflection Based on Linear Polarization Control for Fluorescence-Induced Diagnostic Evaluation

**DOI:** 10.3390/diagnostics12081990

**Published:** 2022-08-16

**Authors:** Sangyun Lee, Kicheol Yoon, Jungmin Kim, Kwang Gi Kim

**Affiliations:** 1Department of Health and Safety Convergence Sciences, Korea University, 145, Anam-ro, Seongbuk-gu, Seoul 02841, Korea; 2Department of Health and Environmental Convergence Sciences, Korea University, 145, Anam-ro, Seongbuk-gu, Seoul 02841, Korea; 3Medical Devices R&D Center, Gachon University Gil Medical Center, 21, 774 Beon-gil, Namdong-daero, Namdong-gu, Incheon 21565, Korea; 4Department of Biomedical Engineering, College of Medicine, Gachon University, 38-13, 3 Beon-gil, Dokjom-ro 3, Namdong-gu, Incheon 21565, Korea; 5Department of Biomedical Engineering, College of Health Science, Gachon University, 191 Hambak-moero, Yeonsu-gu, Incheon 21936, Korea; 6Department of Health Sciences and Technology, Gachon Advanced Institute for Health Sciences and Technology (GAIHST), Gachon University, 38-13, 3 Beon-gil, Dokjom-ro, Namdong-gu, Incheon 21565, Korea

**Keywords:** tumor diagnosis, fluorescence agent, surgical microscope, specular reflection, linear polarized filter

## Abstract

The primary goal of cancer surgery is to completely eliminate tumors. A real-time diagnostic method uses a fluorescence contrast agent and a surgical microscope to assess the status of tumor resection and the patient’s blood circulation. The biggest problem in imaging diagnostics using a microscope is the specular reflection phenomenon. While observing a lesion, the observation field may be obstructed due to specular reflection, making it difficult to obtain accurate results during the diagnostic process. Herein we propose a method to reduce specular reflection during tumor diagnosis by introducing a linearly polarized filter for a surgical microscope system. The method of angular direction adjustment of the filter ensures that only the horizontally polarized light passes through it, thereby obstructing the specular reflection. As a result of removing specular reflection, clear images were obtained at 90° and 270°. This experiment was conducted using phantoms and animals. Our results prove that the proposed method can be applied to imaging cameras used in internal medicine, surgery, and radiology for diagnosis.

## 1. Introduction

Invasive tumors have a high probability of recurrence owing to the penetration into surrounding tissues. The greatest goal of cancer surgery is to completely excise tumors to increase the five-year survival rate of patients and prevent further recurrence of these invasive tumors. During surgery, it is crucial to eliminate the damage to the blood vessels distributed within the tumor in addition to obliterating the tumor [1].

In particular, it is difficult to distinguish the boundary between the tumor and blood vessel with the naked eye owing to their similar shape and color [2]. Therefore, a surgical fluorescence microscope, which uses a fluorescence contrast agent, is used to distinguish tumors from blood vessels [3,4,5,6,7].

This method can separately identify the location of the tumors and blood vessels. Thus, surgical fluorescence microscopy can enable the removal of tumors while tracking the damage done, if any, to the nearby blood vessels [8]. However, the radiation from the light emitting diode (LED), which is used as the light source in the microscope, introduces specular reflections that can overlap at the location of the lesion and thereby obscure the observation field. Therefore, the occurrence of specular reflection interferes with lesion observation and is a major factor hindering accurate diagnosis [9].

To address this issue, several methods for suppressing specular reflection have been previously reported; however, research cases are still lacking [9,10]. Specular reflection is commonly minimized by adjusting the camera’s orientation angle in medical practice. There are two available methods for attaining the polarization of the camera: first, by irradiating the focal point of the LED in various directions, and second, by controlling the focal length of the LED beam [11]. However, it is not easy to adjust the angle of the beam irradiation direction and focal length of the LED in the medical field [12]. In addition, the best imaging condition for the suppression of specular reflection is achieved via accurate detection of specular reflection. In this method, specular reflections are generated in the process of spatial transformation of the red-green-blue (RGB) color function for the brightness and color of the camera, and the detected specular reflection removes the reflection region by performing a deconvolution analysis in the wavelength band [13].

The color image lost due to specular reflection removal uses a separate function to restore the lost color [13]. This reflected wave removal process requires a large amount of data, calculations, and complex mathematical operations [13]. The reference image must be trained in advance for collecting frames though continuous photography [14]. Here, the interval between the actual photography frame and the reference training image interval is compared, and further, the sum and difference are calculated. Therefore, by removing the image for the calculated difference, the corresponding reflected specular can be eliminated [14]. However, this method requires a complex collection process for learning the lesion, normal tissue, and reference image data [15].

Another method has been reported, which involves the use of a polarizing filter in the camera. Alternatively, a sensor can be used to automatically detect changes in the image and calculate only the RGB values of the image [16,17,18,19]. However, the use of a polarizing filter reduces the photographic radius resulting from material loss and provides a dark image.

The RGB value detection method does not consider the image loss and distortion due to material loss of the filter. Methods for suppressing specular reflection of photographed data require further research to overcome the complex process of data collection and further mathematical analysis [16,17,18,19].

In the gradient image transmission method, the captured image may be set to have a high resolution and the specular reflection may be set to have a low resolution. Thus, the above method can be used to perform gradient mapping via pixel optimization [20], similar to previously applied methods [21,22]. This method induces gradient mapping so that only high pixels can be selected. Selected pixels provide an image with effectively reduced specular reflection [21,22]. This method requires complex analysis and algorithm design [20].

In this study, the proposed surgical fluorescence microscopy system aimed at eliminating specular reflection using a linearly polarized filter (LPF) is presented. To remove specular reflection, an LPF is rotated with respect to the angle of vertical and horizontal polarization. The polarization wave is set in the horizontal direction via rotation control, and the rotation angle of the filter becomes 90°. Therefore, a 90° difference between the angle of vertical and horizontal polarization is induced.

## 2. Analysis of Fluorescence Emission and Specular Reflection

A surgical fluorescence microscope system is used to acquire images by injection of a fluorescence contrast medium, irradiation with an external light source, and imaging, as shown in Figure 1a. The blood vessels or tumors can then be identified through their color [23,24,25].

As shown in Figure 1b, the purpose of using a fluorescent contrast agent is to observe the blood circulation status of blood vessels and to remove the tumor through fluorescent staining. Additionally, follow-up of the status of the retained tissue after tumor resection can be performed [23,24,25].

Surgical microscopes are classified into fixed type, pen type, and hand-held type, as shown in Figure 2a [8,20]. These devices are used for observation imaging during the surgery. These structural features are built in the LED to brighten the dark field of view around the imaging camera, as shown in Figure 2b, and the observation method using a fluorescent contrast agent secures the field of view to effectively observe the tissue from the operating microscope, whereas the light from the LED brightly illuminates the tumor [1,2,12,26]. In addition, since the fixed type of probe is large and heavy, it is mounted on the arm and fixed. Therefore, movement in the operating room is not free and the range that can adjust the camera angle is limited. On the other hand, the hand-held and pen type are miniaturized so that the probe can be held by hand. Because of its good mobility and the small size of the probe, it has the advantage of being able to easily access areas that are difficult to image in a fixed device.

However, specular reflection occurs during observation imaging, which is attributed to the effect of tissue density and mucosal moisture status.

According to Snell’s law, for the light (η_1_) irradiated from the LED (as shown in Figure 3a), a part of the light (η_2_) has a refraction angle (θ_2_) which is absorbed by the tissue, and the remaining light is reflected (η_r_) [27]. Therefore, η_1_ is equivalent to η_r_ (η_1_ = η_r_ (θ_1_ = θ_r_)) [28]. Hence, the reflected specular (η_r_) is generated in the form of diffused reflection, as shown in Figure 3b [29]. They are reflected along various directions depending on the angle of the irradiated light.

More specifically, according to Snell’s law, light reflection is caused by an incident wave, a transmitted wave, and a reflected wave. At this time, in Figure 4, when light (η_1_) incident through the medium (air) of m_1_ is transmitted (T) through the medium (tissue) of m_2_, the transmitted light (η_2_) is expressed by Equations (1) and (2) As shown, it has a transmitted wave (T) of an electric field (E) and a transmitted wave (T) of a magnetic field (H) [30].

In this case, the transmitted electric field is E_T_, and the transmitted magnetic field is defined as M_T_. Therefore, since m_1_ (air) and m_2_ (tissue) have different substances, the electric field (E) and magnetic field (H) have a respective angle of incidence (*θ*_1_) and refractive angle (*θ*_2_) for transmission (T) [30].
(1)$$E_{T}=\frac{E_{T}}{E}=\frac{2cos\theta }{cos\theta +\sqrt{n^{2}-sin^{2}\theta }}$$
(2)$$M_{T}=\frac{E_{T}}{E}=\frac{2ncos\theta }{n^{2}cos\theta +\sqrt{n^{2}-sin^{2}\theta }}$$

Here, the transmitted wave (T) for the electric field (E) and the magnetic field (H) has a reflection angle (*θ*_Γ_) and a reflected wave (Γ), and these electric field (E) and magnetic field (H) are expressed in Equation (3) and as shown in (4), it has an electric field (E_Γ_) and a magnetic field (H_Γ_) for reflection (Γ). Therefore, the electric field (E_Γ_) and magnetic field (H_Γ_) have a reflection angle (*θ*_Γ_) [30].
(3)$$E_{\Gamma }=\frac{E_{\Gamma }}{E}=\frac{cos\theta -\sqrt{n^{2}-sin^{2}\theta }}{cos\theta +\sqrt{n^{2}-sin^{2}\theta }}$$
(4)$$M_{\Gamma }=\frac{E_{\Gamma }}{E}=\frac{-n^{2}cos\theta +\sqrt{n^{2}-sin^{2}\theta }}{n^{2}cos\theta +\sqrt{n^{2}-sin^{2}\theta }}$$

At this time, the LED light irradiated to the air layer (m_1_) is absorbed by the tissue fluorescence contrast agent (m_2_), and the fluorescence contrast agent emits fluorescence as shown in Figure 4 due to a chemical reaction. If the tissue in which the fluorescent contrast agent is absorbed has a higher density than the air medium (m_1_ < m_2_), a transmitted wave (T) for fluorescence that becomes emission is generated as in Equation (5). Therefore, when the transmitted wave (T) for the emitted fluorescence is photographed by the camera, reflected light (Γ) is generated as shown in Equation (6). So, as shown in Figure 5, the transmitted waves (E_T_ and M_T_) for electric field (E) and magnetic field (H) with angles of incidence (*θ*_1_) and reflection (*θ*_r_) in the range of 0° to 80° for the transmission coefficient are the transverse coefficient approaches 0 and the reflection coefficient approaches 1. However, the reflected waves (E_Γ_ and M_Γ_) for the electric field (E) and magnetic field (H) generated through fluorescence expression have a transverse coefficient of 1 and reflection when the incident angle (*θ*_1_) and the reflection angle (*θ*_r_) are 90°. The coefficient becomes 0 and light reflection is suppressed. Therefore, as shown in Figure 4, E_T_ and M_T_ are captured by the camera, and E_Γ_ and M_Γ_ are reflected back [30].
(5)$$\|T\|=\left(\eta \frac{cos\theta_{2}}{cos\theta_{1}}\right){\Gamma \Gamma }^{*}=\left(\eta \frac{cos\theta_{2}}{cos\theta_{1}}\right)T^{2}$$
(6)$$\|\Gamma \|={\Gamma \Gamma }^{*}={\left|\Gamma \right|}^{2}$$

Subsequently, specular reflections from the LED light are photographed using a camera, and a white blob of light is generated in the image when performing endoscopy, or observing a tumor and blood circulation, as shown in Figure 3c [30]. If such specular reflection occurs in an important lesion area in the image, it may obstruct the observation field of view, thereby introducing a diagnosis error [30]. Therefore, a filter is required to remove such specular reflection.

According to the analysis, Ref. [30] has similar properties as a method for removing light reflection. The characteristic of [30] is the generation process of diffuse reflection and specular reflection by LED in gastroscope or colonoscope. When the LED is irradiated on the surface of the tissue mucosa, light reflection on the LED occurs due to moisture on the mucosal surface. In this case, light reflection includes diffuse reflection and specular reflection. The characteristic of light reflection by LED is mainly white light, and the wavelength of irradiation and the wavelength of reflection are the same. However, the feature of this study is light reflection by fluorescence expression. When the LED is irradiated to the tissue, a chemical reaction occurs in the tissue by the fluorescent contrast agent. At this time, fluorescence is emitted by a chemical reaction in the tissue. Therefore, diffuse reflection and specular reflection correspond to the fluorescence emission wavelength band, not the LED irradiation wavelength band, as shown in Figure 6a, i.e., the diffuse reflection and specular reflection generated by the chemical reaction of the fluorescent contrast agent as shown Figure 6b [31].

For this reason, the results of this study are analyzed for light reflection caused by light reflection and fluorescence emission by LED irradiation, and a method to effectively reduce light reflection is studied. Therefore, in the diagnosis process, the visual field of the lesion is blocked by light reflection to reduce the phenomenon that interferes with diagnosis. 

In [30] LED irradiation reflects the light to the LED irradiation by moisture on the surface of the mucous membrane. This light reflection has a white light wavelength band. However, since the fluorescence expression observes the flow condition of the blood vessel, the light reflection due to the fluorescence expression obscures the field of view of the minute blood flow condition [31]. That is, since light reflection due to fluorescence expression appears in the form of a dot, it blocks the field of view for a minute blood flow condition, which can be very difficult to diagnose. Therefore, we propose a method to solve such fluorescence light reflection.

There are several types of filters used: ultraviolet (UV) filter, skylight filter, polarizer (PL) filter, and neutral density (ND) filter [32,33]. The UV and skylight filters are mainly used to reduce specular reflection caused by UV rays [32]. The PL ND filters are used to reduce the diffused reflection and the specular reflection by uniformly reducing the amount of light to balance the entire color images [33]. However, these filters are mainly suitable for imaging in bright natural environments. The PL filter exhibits excellent performance in terms of specular reflection reduction, and consequently, is very widely used [33]. However, the background becomes dark when using a filter. This was first observed in the phantom experiment and was also published in subsequent papers [9]. Therefore, if a PL filter is used in a state that only relies on LED in a dark body, specular reflection is effectively reduced. However, the applicability of PL filter is limited in securing the field of view of the lesion because of the dark image [9]. Thus, using a PL filter makes it difficult to observe the lesion. In addition, further research needs to be conducted to effectively remove specular reflection in a dark environment and observation of the lesion in a bright field of view is necessary.

## 3. How to Eliminate Occurrence of Specular Reflection 

The occurrence of specular reflections indicated by the white blobs in the microscopy images inhibit the accurate identification of surgical sites and lesions. Introducing a filter controls the polarization of the incident light, thereby filtering the light which was previously observed as specular reflection. Figure 7 depicts this elimination phenomenon, achieved using a filter [30].

An image obtained using a camera includes both the vertically (V_in_) and horizontally polarized light (H_in_) [19,20]. If an image is taken from the camera without a polarization filter, the images corresponding to the vertical (V_in_) and horizontal polarization (H_in_) are incident on to the camera as shown in Figure 8a [34,35,36].

V_in_ and H_in_, incident through the camera, exhibit the property of diffuse reflection. In addition, the actual image has the diffuse reflection property of the white light source as shown in Figure 8b.

To suppress the diffused reflection, it is necessary to remove the vertically polarized light (V_in_). This can be achieved by controlling the rotation angle between the two filters, (F_1_) and (F_2_), as shown in Figure 9a. F_1_ is fixed and F_2_ is rotated relatively such that the specular reflection is eliminated effectively.

When F_1_ is fixed and the rotation direction angle (r_1_) is 0° as shown in Figure 10, V_in_ and H_in_ incident on the filter have vertical and horizontal polarization at P_0_, respectively. At this point, when V_in_ and H_in_ pass through F_1_, both V_in_ and H_in_ change to horizontal polarization at P_1_.

If the direction angle (r_1_) of the F_1_ filter becomes 0° and the direction angle (r_2_) of the F_2_ filter is changed, the intensities of V_in_ and H_in_ passing through the filter are equal to those calculated by Equations (7)–(10) in P_3_.

The polarization effect is also achieved, and the specular reflection will be sufficiently eliminated. Since the specular reflection orientation is rotated, the specular reflection intensity (*I_ref_*) with respect to the direction angle (r_2_) of the filter corresponding to the 3/4 (π/4) when the phase angle (θ) is 60° will be decreased [36].
(7)$$I_{ref}=I_{Ep}cos^{2}\theta$$
(8)$$2cos=2I_{Ep}cos^{2}\theta$$
(9)$$1/2I_{Ep}cos^{2}\theta$$
(10)cos90°

For example, the intensity of specular reflection (*I_ref_*) of filter (F_1_) is fixed at 0°, and that of filter (F_2_) is 90°. Then, the phase angle (*θ*) of the specular reflection passing through the filter (F_2_) is equal to a difference of 90° (*θ* = 90°). Therefore, V_in_ is not passed, and H_in_ is passed. The specular reflection intensity *I_EP_* for P_3_ is reduced by more than half compared to that of P_2_. Here, if the *I_EP_* is 8 mW/cm^2^, the rotation angle *θ* of F_1_ is 50°, and when the rotation angle *θ* of F_2_ is 20°, the *I_ref_* is expressed by Equation (11) and is equal to 6 mW/cm^2^. Therefore, when *I_EP_* is 50 mW/cm^2^ (*θ* = 0°), *I_ref_* becomes 0 mW/cm^2^ (*θ* = 90°), as shown in Table 1. Simultaneously, when the rotation angle *θ* of the filter (F_1_) is 0°, the rotation angle *θ* of the filter (F_2_) is changed from 0° to 360°.
(11)$$\begin{array}{l} I_{ref}=I_{Ep}cos^{2}\theta =\left(8\times 10^{-3}\right)cos^{2}\left(50{}^{\circ}-20{}^{\circ}\right)\\ \;\;\;\;\;\;=\left(8\times 10^{-3}\right)cos^{2}\left(30{}^{\circ}\right)\\ \;\;\;\;\;\;=\left(8\times 10^{-3}\right){\left(\frac{\sqrt{3}}{2}\right)}^{2}=60\;\mathrm{mW}/{\mathrm{cm}}^{2} \end{array}$$

Figure 11 presents a graphical representation (sinusoidal function) of the data listed in Table 1. Thus, rise and fall of the sine wave indicates maximum retention and complete elimination of the specular reflection, respectively.

In the end, the intensities for *I_EP_* and *I_ref_* are the same and their intensities have an equilibrium relationship with each other. If the filter rotates up to 10° to 80° during the rotation of the second polarizing filter F_2_, *I_EP_* and *I_ref_* are out of equilibrium.

If the rotation axis r_2_ of the second filter F_2_ is 90°, *I_EP_* and *I_ref_* (the difference between filter F_1_ ≠ F_2_ = 90°) exhibit a cross relationship. So, *I_EP_* has the maximum value (*I_EP_* = 1) and *I_ref_* has the minimum value (*I_ref_* = 0). Therefore, the intensity of specular reflection is lost, and the image quality is increased. Eventually, *I_EP_* and *I_ref_* will have the same value (*I_EP_* = *I_ref_*), and this phenomenon is repeated. To summarize, the specular reflection is maximally suppressed when the orientation of F_1_ and F_2_ both is 90°. As a result, the polarized wave (p_3_) that has passed through F_2_ outputs only the polarized wave (H_in_ = 0) with reduced specular reflection.

The intensity of light loses its maximum value, and eventually, begins to decrease slowly. When the rotation angle (r_2_) is 90° (270°), the intensity of specular reflection (*I_ref_* = 0) becomes 0.

## 4. Experiment Configuration and Results

### 4.1. Experiment Configuration

In this study, we apply two different methods to eliminate specular reflection. The first method requires the use of a phantom, and the second method involves animal testing to obtain reliable results.

#### 4.1.1. Phantom Production

To obtain the results for the specular reflection cancellation effect, the fabricated phantom was capable of indocyanine green (ICG) fluorescence expression. For phantom production, latex rubber solution, saline, and fluorescent contrast agent ICG are mixed, as shown in Figure 12.

To prepare the fluorescence contrast agent, ICG powder (25 mg) is diluted with saline (10 cc). Here, the concentration of ICG is 2.5 mg/mL. Latex rubber solution with a molecular weight of 774.98 g/mol has a volume of 1 vial (50 mL). The solution was prepared such that its molar concentration (amount of substance per unit volume of solution, M) was equal to 3.23 nM, using Equation (12).

ICG for a concentration of 2.5 mg/mL adjusts the dilution value by diluting it with a rubber solution having a volume of 50 mL. Then, the ICG adjusts the injection volume value of the syringe (0.2–0.4 cc). Therefore, the ICG, rubber latex liquid, and diluted molar concentration are classified into 19 μM, 20 μM, and 30 μM units, respectively [37].
(12)$$\mathrm{molar}\;\mathrm{concentration}=\frac{I/W}{L}\left[\mathrm{mol}\right]$$

The mixed liquids were each injected into aluminum weighing dishes, heated to 100 °C, and left for 60 min. As a result, a solid phantom was manufactured as shown in Figure 12.

#### 4.1.2. Phantom Reagent Production

A fluorescent contrast agent, namely, ICG, was used to perform the phantom and animal testing experiments for verifying the specular reflection removal as shown in Figure 13. An ICG concentration of 2.5 mg/mL was attained by diluting 25 mg of ICG powder with normal saline (10 cc). Additionally, the ICG (2.5 mg/mL) was diluted with 25 mL of normal saline to ensure a dilution of 10 times or more. ICG that satisfies the dilution criterion generates an injection volume of 0.2–0.4 cc using a syringe, and the ICG (dilution) induces injection into the phantom or vein (syringe: 0.2–0.4 cc).

Experimental results using the phantom were tested on small animals (rat) to ensure the reliability of the results. Animal experiments were tested at the Laboratory Animal Center of the Lee Gil-yeo Cancer and Diabetes Research Institute in Korea, and the IACAU number is LCDI-2017-0050. We were approved by the Animal Ethics Committee of the Institutional Review Board. The animal type, age, weight, and strain were rat (male), 8 weeks old, 240 g, and Sprague Dawley (SD), respectively. The ICG fluorescence contrast medium (concentration: 0.005 mg/kg) was injected intravenously (0.2–0.4 cc), as shown in Figure 14.

The injected ICG is bound together with plasma protein, and it circulates through blood vessels.

An ICG with an injection volume of 0.2–0.4 cc is irradiated with an external light source such as LED or laser. It exhibits fluorescence activity within a wavelength range of 805 nm–860 nm. If the infrared (IR) band of 803 to 860 nm is photographed using a near infrared (NIR) camera, the vein is expressed in a color suitable for the IR band, and eventually, blood vessels based on ICG fluorescence expression can depict the blood flow phenomenon in the IR band [38].

#### 4.1.3. Experiment Device Configuration

To ensure effective specular reflection removal, the experimental device shown in Figure 15 must be used. A schematic diagram of the experimental setup consists of a camera, an LPF, and an LED, and the results are obtained using the ICG phantom.

Subsequently, the imaging and observation of the removal of specular reflection was performed by using a self-made hand-held type of fluorescence surgical microscope.

A hand-held fluorescence surgical microscope has the following characteristics. The component devices are connected by an LED, IR camera, LED light intensity control function, ON/OFF switch, a power supply, and external monitor.

LPF_1_ is connected to the front end of the LED, and the rotation angle (θ) is 0°. In addition, an LPF (cut on wavelength: 800 nm @ thorlabs FELH0800) that can pass only the emission wavelength (820 to 860 nm) and two LPFs (LPF_1_//LPF_2_) are connected to the front of the camera head. Here, the rotation angle θ of the LPF_1_ is 0° (unpolarized) and that of LPF_2_ can be rotated from 0 to 360° (unpolarized or polarized). The polarization state for specular reflection requires the rotation angle (θ) of the LPF (F_2_) to be set to 90° (polarized).

The working distance (WD) from the camera and LED to the phantom is 3 cm, and the angle (θ) between the LED and the camera to focus the beam is about 15°.

For accurate imaging, the self-made hand-held type of fluorescence surgical microscope was fixed in position using a clamp device and was manufactured using the 3D printer technology. The main parameters of the LED, camera and filters used for the experiment are presented in Table 2.

For using an LPF (Corning Polarcor™ Glass Polarizer), the wavelength range is set to 600 to 1100 nm. The diameter and thickness of the filter lens are 5.0 and 0.5 mm, respectively. The transmittance is 66%, the insertion loss is 1.8 dB and the angle of incidence beam 5.0°.

### 4.2. Experiment Results

Figure 16 shows the experimental results obtained using the ICG phantom. Here, the yellow marker indicates specular reflection which was observed in images obtained before using the filter. When the rotation angle (θ) of the filter connected to the LED rotated the filter in units of 0° to 360°, a specular reflection of reduced intensity can be observed. When the rotation angle (θ) is 90 and 270°, the specular reflection is sufficiently reduced, and its intensity is analyzed differently depending on the rotation angle (θ) for other results. To obtain reliable results, this imaging testing was performed three times (without LPF and with two different LPF), and were found to produce consistent results.

LPF_3_ is 0° and the relationship is parallel to LPF_1_. Although the existence of LPF_3_ is not required in Malus’ law, it is necessary when designing a system. The reason is that when LPF_1_ and LPF_2_ are 0° and 90°, respectively, light reflection is weakly suppressed as shown in Figure 17. At this time, if FPF_3_ (0°) is inserted, light reflection is strongly suppressed as shown in Figure 14 because LPF_1_ and LPF_2_ change into a structure (P_3_) that overlaps as shown in Figure 9b.

This study proposes a method to remove light reflection and light reflection generated from LEDs of an endoscope. Of course, when laser is used, light reflection is analyzed to occur steadily [39]. In addition, it is estimated that the optical sources (LEDs and lasers) for all, incoherent and coherent, will cause light reflection, so the light reflection will result in both diffuse reflection and special reflection, as shown in Equations (1)–(6) and Figure 17. This study analyzes light reflection (diffuse reflection and special reflection) by LED irradiation and light reflection (diffuse reflection and special reflection) generated by chemical reaction of fluorescent contrast agents when irradiating LEDs and suggests a method to effectively remove such light reflection. When the LED is irradiated, the light without filter is strongly reflected in the center of the phantom, and the area around the phantom is diffuse reflection, as shown in Figure 17. At this time, if LPF_2_ (Figure 15) is rotated (0°–180°) in Figure 18, it can be seen that light reflection is suppressed (98.9%) when the angle of rotation (LPF_2_) is 90° as shown in Figure 18.

In the case of the laser, as illustrated in Figure 18, the light filter generates wide and strong reflections in the center of the phantom. However, if LPF_2_ (Figure 15) is rotated (0°–180°), light reflection is suppressed (97%) when the angle of rotation (LPF_2_) is 90° as shown in Figure 18. The reason is that if the light reflection intensity (I_EP_) becomes cos290° as shown in Equations (7)–(10), when the rotation angle of F_1_ is 0° as shown in Figure 10, F_2_ has an orthogonal relationship, so that the magnetic field (M_T_) is offset and the electric field (E_T_) disappears inversely.

In Figure 19, the captured images were simulated using the region of interest (ROI) program. In the simulation results, when the rotation angle θ of the filter is 0°, the specular reflection occurrence position is indicated by a yellow point.

In this case, the intensity (intensity of light) corresponds to 0 in the area where specular reflection could not be observed in the histogram. However, since most of the specular reflection occurs in the phantom, the intensity of specular reflection is no longer 0. Therefore, the intensity of specular reflection has a value of 25 to 294, and the maximum intensity corresponds to 294. However, when the rotation angle θ of the filter is 90°, the intensity of specular reflection changes to 0.

This is recommended as a method for fluorescence diagnosis in the future and is expected to be highly utilized. We also believe that these research methods will be fully applicable to the healthcare sector through product design and clinical trials in the future.

In order to numerically observe the removal state of light reflection in the resulting images of Figure 16 and Figure 19, it is necessary to obtain the speckle noise contrast evaluation result of the phantom as shown in Figure 20. At this time, the beam homogeneity should be observed. In the result, the red waveform is the speckle noise condition without using the LPF, and the blue waveform is the result of the light reflection removal condition using the LPF.

In the results, the phantom on the right is the phantom before and after using the LPF as shown in Figure 19, and the left side is the result of speckle noise conditioning for the phantom before and after using the LPF.

At this time, light reflection occurs before using the LPF, but it can be confirmed that the light reflection is removed when the LPF is used.

Therefore, the probability density (*p_EP_* (*I**_θ_* = 0°)) of the speckle noise before and after using the LPF is the same as the Equations (13) and (14). At this time, the parameter (*τ*) is 0.21 as in Equation (15) [40].
(13)$$p_{EP\left(I_{\theta =0{}^{\circ}}\right)}=\frac{I_{ref}}{2\tau }e^{\left(\frac{-I_{EP}^{2}}{2\tau }\right)}\;$$
(14)$$p_{EP}\left(I_{\theta =90{}^{\circ}}\right)=\frac{I_{ref}}{2\tau }e^{\left(\frac{-I_{EP}^{2}}{2\tau }\right)}\;$$
(15)$$\tau =\frac{1}{2}\left(1-\frac{\sqrt{3}}{3}\right)=0.21$$

Here, *p_EF_* (*I**_θ_* = 0°) without LPF is 0.0057 and *p_EF_* (*I**_θ_* = 90°) with LPF is 0.0000038. The intensity of light reflection (*I_EP_*) for the results obtained from these images (see Figure 16 or Figure 19) is 100 for the imaging frequency presented in Equations (16) and (17).
(16)$$I_{EP\;\left(\theta =0{}^{\circ}\right)}=I_{ref\left(\theta =0{}^{\circ}\right)}+I_{EP}\theta \left[\frac{1+sin\theta \cos\left(f+\theta \right)}{2}\right]$$
(17)$$I_{EP\;\left(\theta =90{}^{\circ}\right)}=I_{ref\left(\theta =0{}^{\circ}\right)}+I_{EP}\theta \left[\frac{1+sin\theta \cos\left(f+\theta \right)}{2}\right]$$

At this time, the *I_EF_* (*θ* = 0°) when the LPF is not applied in the *I_EP_* of 50 mW is 49.0 mW. However, the *I_EF_* (*θ* = 90°) when the LPF is applied at an *I_EP_* of 50 mW is 0.76 mW. Combining these results, according to the Nyquist sampling law, when the intensity of light reflection removal (*I_EP_*) is less than twice the intensity of light reflection (*I_ref_*) (*I_EP_* ≤ 2*I_ref_*), light reflection is reduced [41].

Figure 21 shows the results of specular reflection removal experiments using rats. From the figure, the experimental results show the state of blood flow through ICG fluorescence.

The samples were irradiated with LED mounted on a self-produced hand-held fluorescence surgical microscope and image data were collected through an IR camera. Then, the results of the imaging experiment were observed through an external monitor. When the LPF was not applied, specular reflection occurred, but when the rotation angle (*θ*) of the filter was adjusted from 0° to 360° using the LPF, the results were consistent with those obtained when the rotation angle was 90° and 270°. Thus, it was confirmed that the reflection was sufficiently removed. The same results as those of the experiment using the phantom can be confirmed through animal experiments.

In the experiment process, the imaging test is separated into two sections, which are divided without LPF and with LPF. The without LPF parts are generation of light reflection imaging (diffuser and specular), and the imaging results are composed of vein with colon, liver (portal and hepatic), derma, and tail, respectively. We observed the blood vessel flow with florescence emission using ICG substance in the vein. Then, the imaging results are generated the light reflection condition (see Figure 21 of yellow marker). Thus, the imaging results are necessary for removal of light reflection. Figure 16 (see under the part) shows the removed light reflection in the imaging results, To remove the imaging of light reflection, a handmade pen type probe is used for LPF which is observes the light reflection removal condition (diffuse and specular), and the observation of the organ locates the vein with colon, liver (portal and hepatic), derma, and tail, respectively. This can also be enough for suppression of the light reflection. In addition, the rotation angle of 90 degrees and 270 degrees are excellent for the removal of light reflection.

## 5. Conclusions

During clinical diagnosis, the orientation angle or brightness of LEDs is adjusted to reduce specular reflections from the captured images. However, if the camera direction angle is adjusted, the lesion in the captured image may be distorted or the shape observation angle may be changed. In addition, if the LED brightness is adjusted, the captured image is observed with a dark background. Therefore, the photograph is corrected using image processing, which takes time and causes inconvenience in the operation.

A clear image for rapid and accurate diagnosis must be provided, and specular reflection must be removed to secure the observation field of the lesion. This study proposes a method to effectively reduce the specular reflection of images generated in a surgical glare microscope system, wherein the polarization of the incident light can be effectively controlled using the rotation angle of the filter. The filter allows only the horizontally polarized light to pass through.

This study aimed to secure the imaging results for the specular reflection removal effect using a phantom, and small animal experiments were conducted to obtain high reliability of the phantom experimental results. The results of these experiments confirmed a decrease in the occurrence of specular reflection.

The proposed method eliminates specular reflection in real time in the operating room without additional software and correction processing. This method can be applied to all cameras, microscopes, and endoscopes regardless of the size of the filter. In addition, such a method provides an excellent photographed image to the operator. Furthermore, the lesion observation field can be secured through clear imaging results devoid of specular reflection, and the advantage is that rapid and accurate diagnosis results can be obtained. This method should be suitable for application in clinical diagnostic sites in the future, and is expected to be highly practical. Furthermore, we believe this research method can be sufficiently applied to the medical field through product design and clinical trials in the future.

## Figures and Tables

**Figure 1 diagnostics-12-01990-f001:**
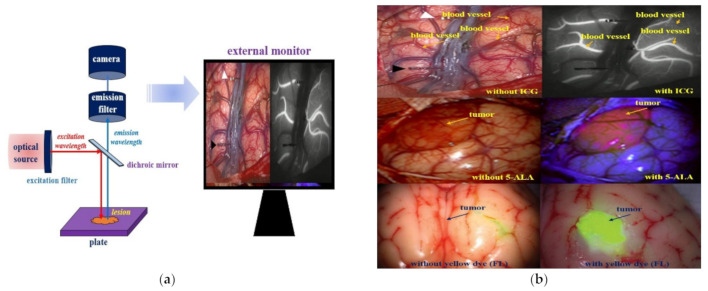
(**a**) Schematic of the fluorescence surgical microscope and external tissue monitoring. (**b**) Observation imaging of tumor during surgery using a surgical microscope.

**Figure 2 diagnostics-12-01990-f002:**
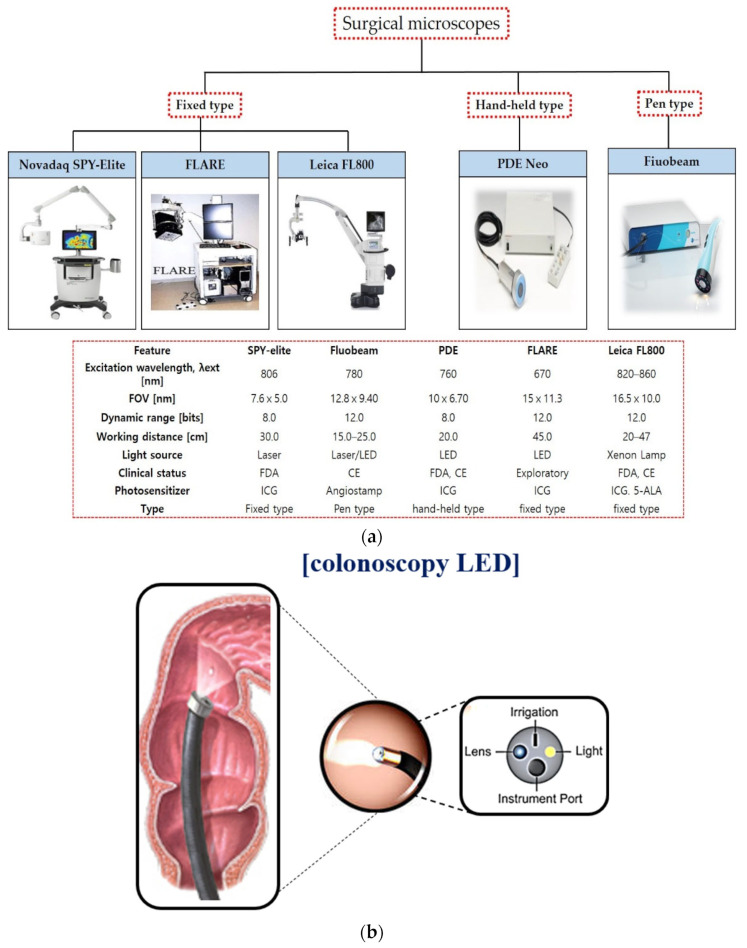
(**a**) Diagnosis process of the surgical fluorescence microscopy system and its (**b**) camera structure.

**Figure 3 diagnostics-12-01990-f003:**
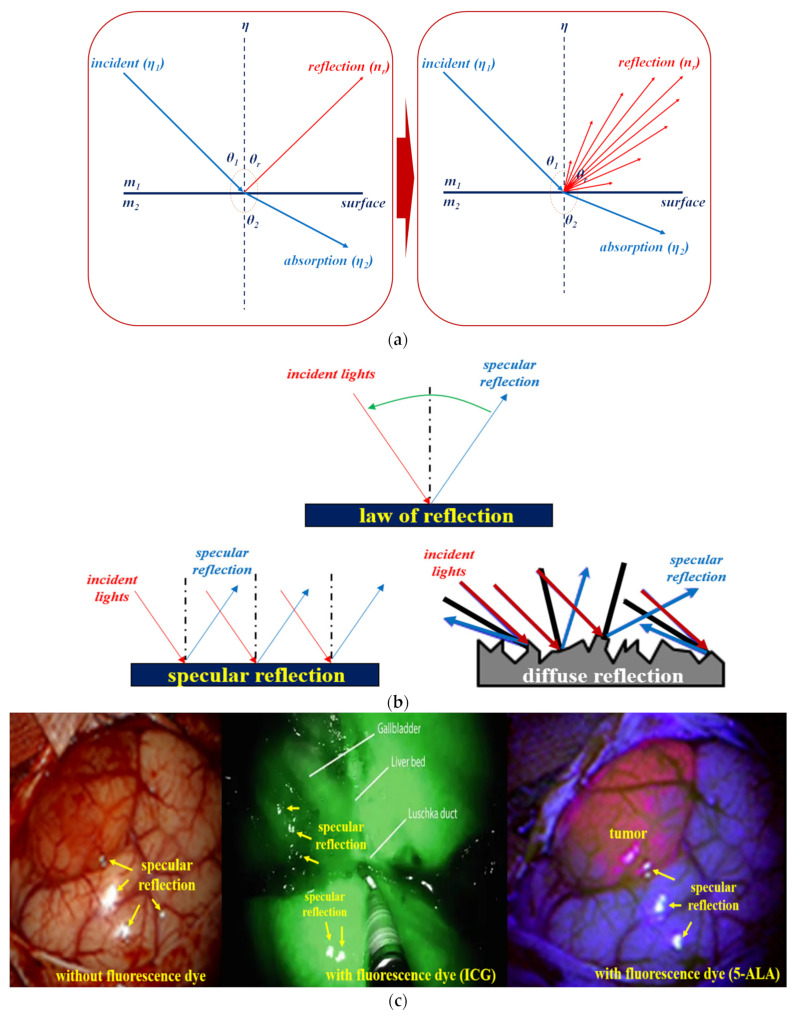
Generation of specular reflection during diagnosis. (**a**) Illustration of Snell’s law. (**b**) Diffused reflection resulting from the varying reflected specular (η_r_). (**c**) Comparison among specular reflection observed for cases without the use of a fluorescent dye and that using a fluorescent dye, namely, indocyanine green (ICG) and 5-Aminolevulinic Acid (5-ALA).

**Figure 4 diagnostics-12-01990-f004:**
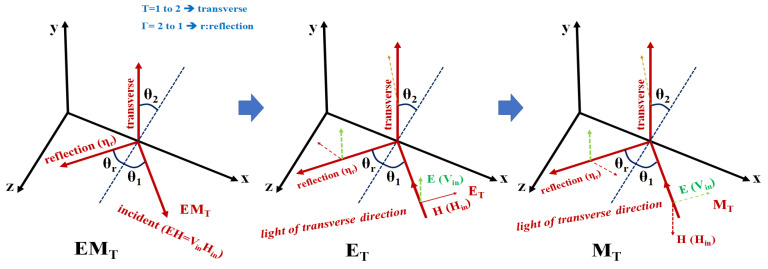
Propagation of electromagnetics for light reflection and transvers.

**Figure 5 diagnostics-12-01990-f005:**
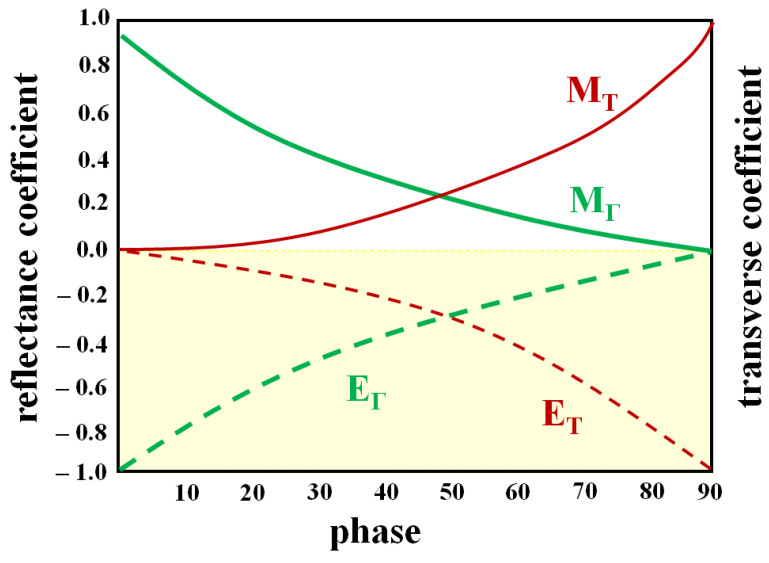
Coefficient of transverse and reflection.

**Figure 6 diagnostics-12-01990-f006:**
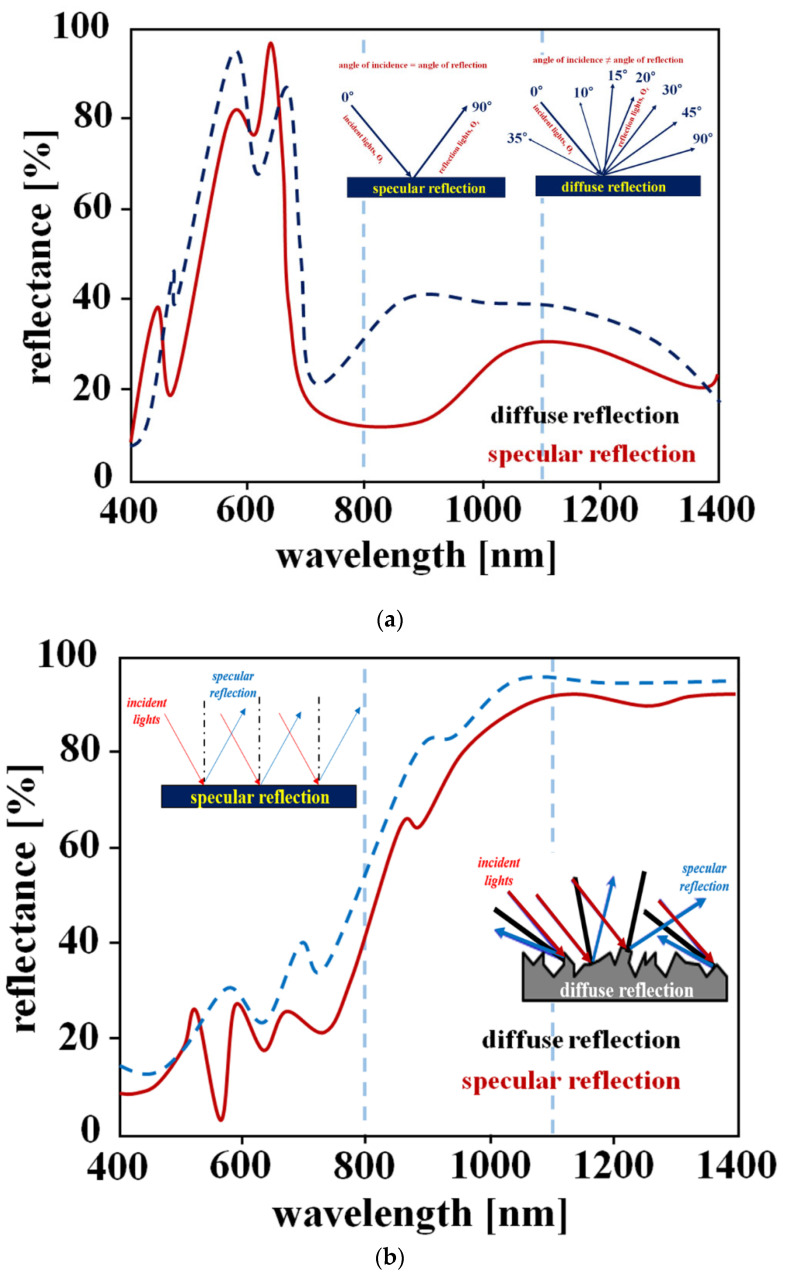
Light reflection coefficients (**a**) LED irradiation (**b**) fluorescence emission.

**Figure 7 diagnostics-12-01990-f007:**
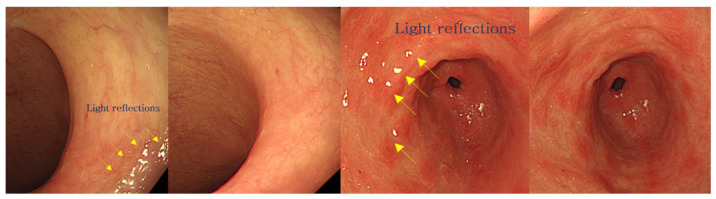
Comparison of images obtained before and after eliminating specular reflection during a colon endoscopy.

**Figure 8 diagnostics-12-01990-f008:**
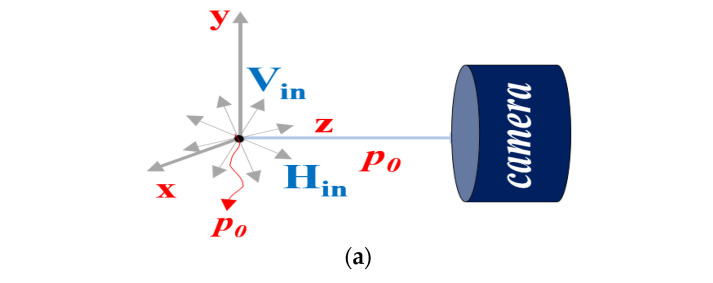
Generation of specular reflection in a camera. (**a**) incident of light polarization (**b**) generation of light reflection phenomenon.

**Figure 9 diagnostics-12-01990-f009:**
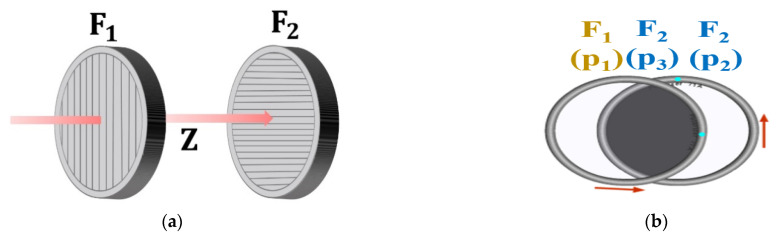
Adjustment of rotation angle using a filter. (**a**) LPF configuration. (**b**) Combination of LPF.

**Figure 10 diagnostics-12-01990-f010:**
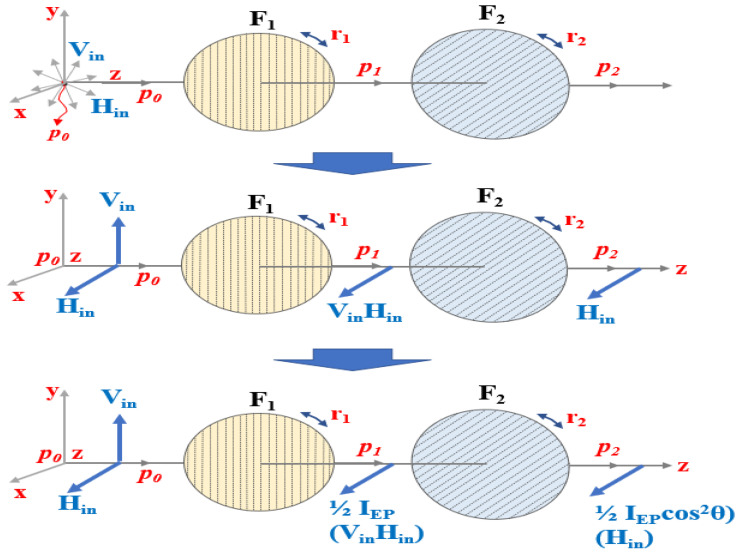
Specular reflection removal attained by controlling the rotation angle of the polarizing filter.

**Figure 11 diagnostics-12-01990-f011:**
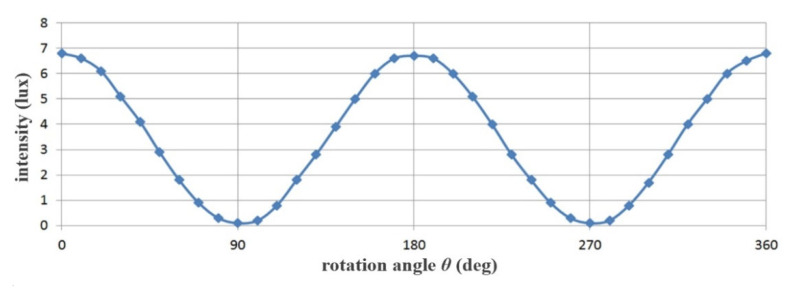
Variation of the specular reflection intensity according to the rotation axis of the filter.

**Figure 12 diagnostics-12-01990-f012:**
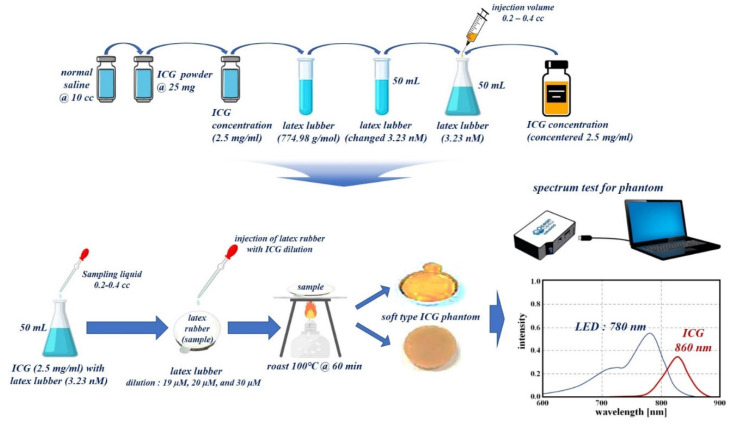
Proposed manufacturing of phantom using ICG and latex solution.

**Figure 13 diagnostics-12-01990-f013:**
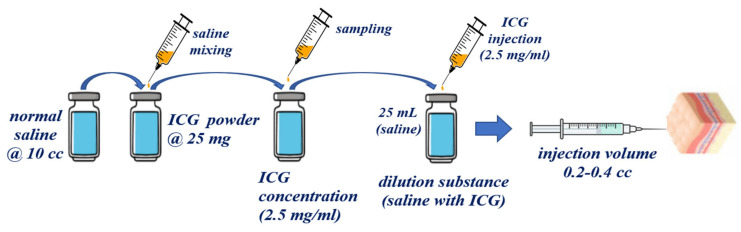
Phantom reagent production procedure.

**Figure 14 diagnostics-12-01990-f014:**
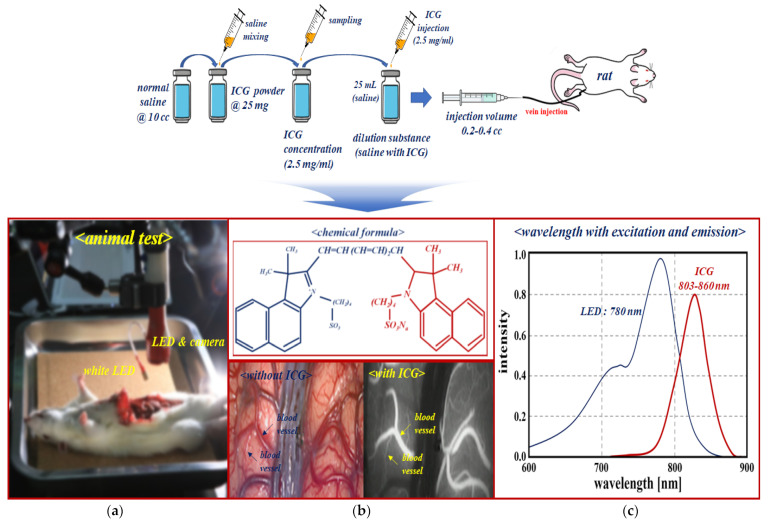
ICG mechanism of action and emission wavelength band (**a**) Fluorescence detection after injection. (**b**) ICG chemical formula. (**c**) ICG fluorescence emission wavelength.

**Figure 15 diagnostics-12-01990-f015:**
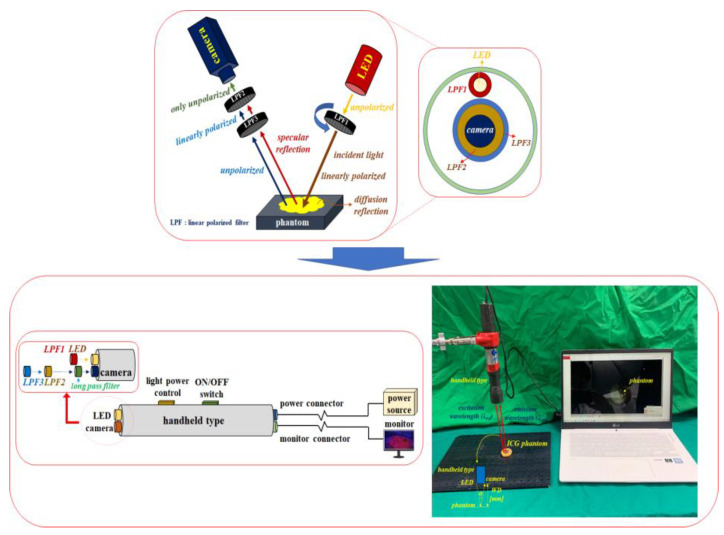
Construction of the experimental device.

**Figure 16 diagnostics-12-01990-f016:**
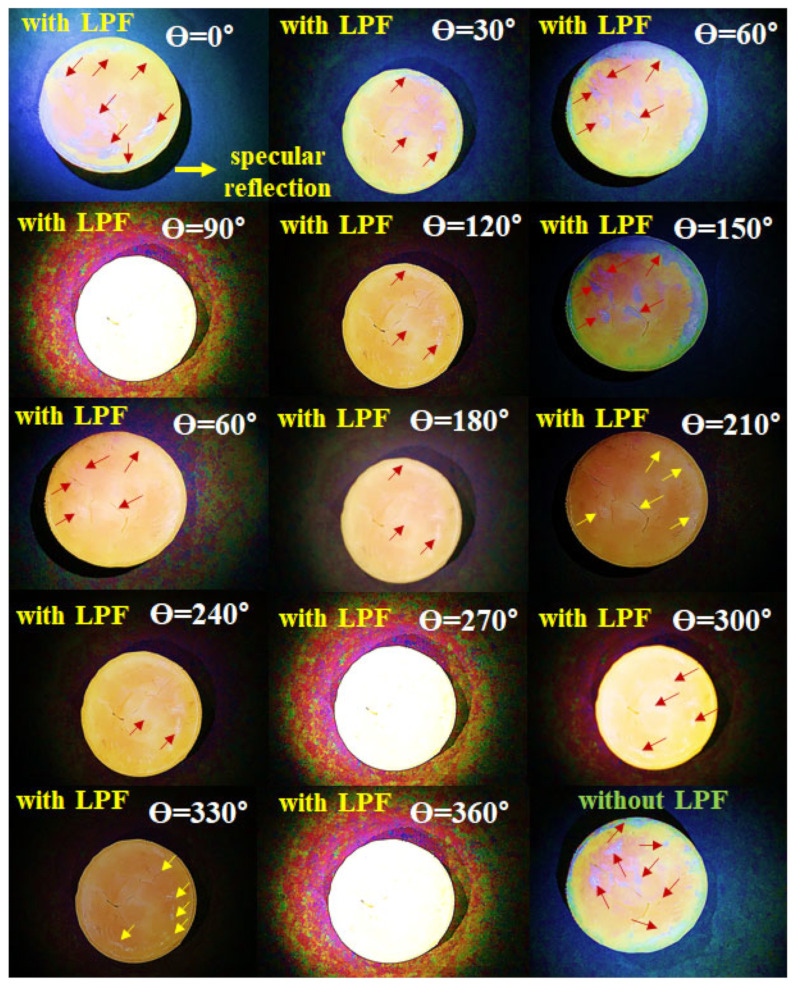
Phantom experiment results (for applications of the LPF_1_, LPF_2_, and LPF_3_).

**Figure 17 diagnostics-12-01990-f017:**
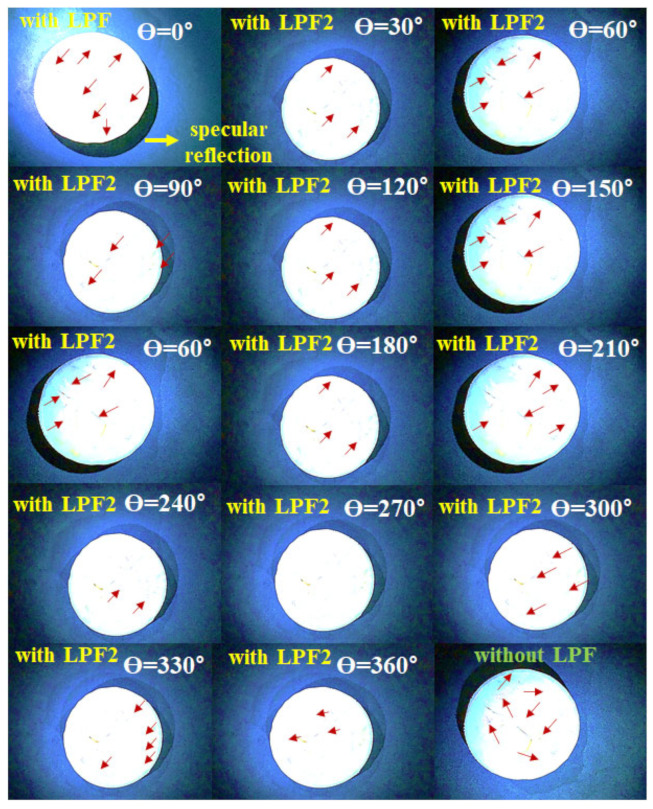
Application of the only LPF_1_ and LPF_2_ (without LPF_3_).

**Figure 18 diagnostics-12-01990-f018:**
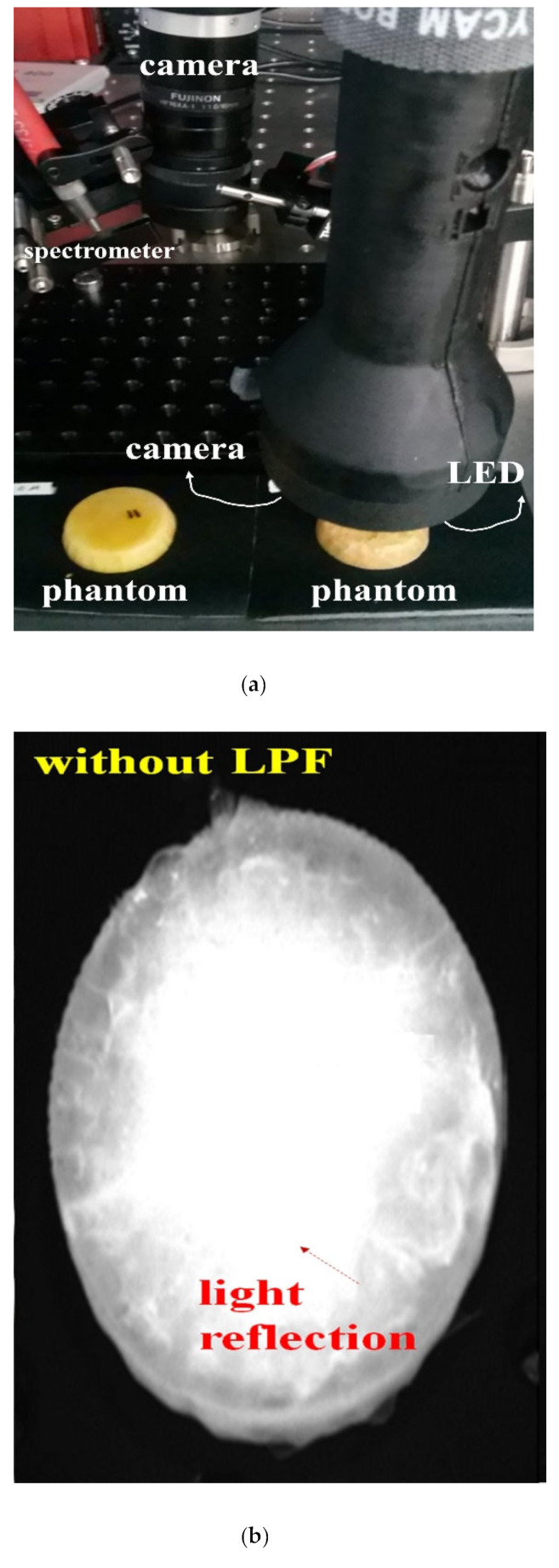
Laser applications (**a**) test configuration (**b**) without LPF (**c**) with LPF.

**Figure 19 diagnostics-12-01990-f019:**
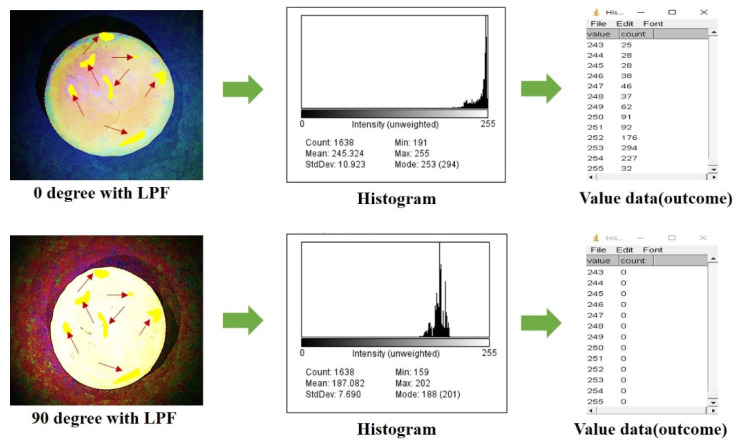
ROI numerical analysis of phantom experiment results for specular reflection removal.

**Figure 20 diagnostics-12-01990-f020:**
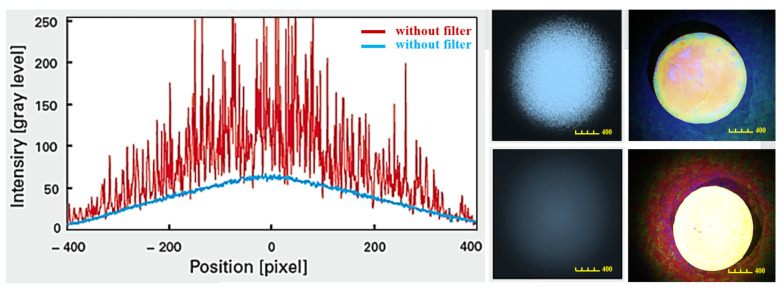
Image test results of speckle noise contrast for light reflection removal through the Figure 19.

**Figure 21 diagnostics-12-01990-f021:**
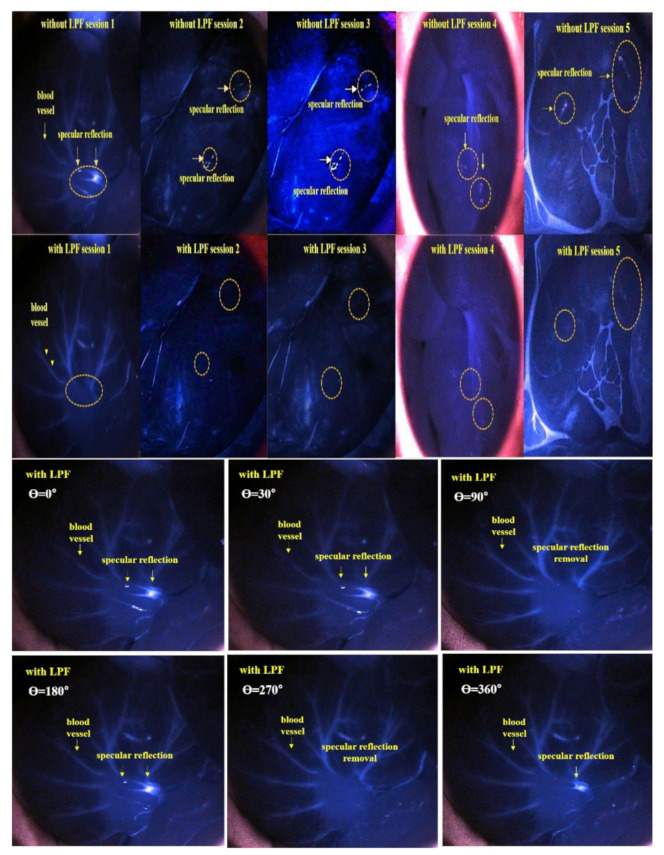
Spectral reflection removal confirmed through small animal experiments.

**Table 1 diagnostics-12-01990-t001:** Analysis of changes in specular reflection intensity according to the rotation angle of filter (F_2_).

Rotation Angle of the Filter F_2_ (θ)	Specular Reflection Intensity [mW/cm^2^]	Rotation Angle of Filter F_2_ (θ)	Specular Reflection Intensity [mW/cm^2^]
0°	50.0	210°	37.5
30°	37.5	240°	12.5
60°	50.0	270°	0.00
90°	0.00	300°	12.5
120°	12.5	330°	37.5
150°	37.5	360°	50.0
180°	50.0		

**Table 2 diagnostics-12-01990-t002:** Experimental device module parameters.

Performance (@ LED)	Parameter	Performance (@ Camera)	Parameter
model	Thorlabs LED 780E	model	SJ-8200
wavelength, λ (nm)	780–785	sensor	CMOS
output power (mW)	18.0	Resolution [P]	1920
current (mA)	100	pixel size [Mpixel]	2.0
voltage (V)	1.75–1.95	frame rete [fps]	30
beam angle of radiation, θ (deg)	10	focal distance	5 mm–infinity
luminous intensity (mrcd)	2500	view angle [deg]	60°

## Data Availability

The data presented in this study are available upon request from the corresponding author. The data are not publicly available because of privacy and ethical re-strictions.

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
