# Peer review of "Reduction of Specular Reflection Based on Linear Polarization Control for Fluorescence-Induced Diagnostic Evaluation"

_diagnostics, 2022, doi:10.3390/diagnostics12081990_

Round 1

Reviewer 1 Report

The authors present measurements showing specular reflection of LEDs as they impact cancer diagnosis due to effects produced on fluorescent/colored imaging systems and how to avoid it with a polarization filter

I feel the approach is quite obvious and I have concerns about the novelty of this work after checking the references 28, 30 and 37. Two of the authors have already published on the subject in this very journal this year; see ref. 30.

The authors focus on study cases and not on accumulating huge date to generate datasets that can lead to rigorous quantification of the specular refection compensation via the polarization.

Pen or handheld devices are mentioned in the text (line 110), but images in the table of Figure 2 shows large devices instead. So the phrase in lines 110-111 is incorrect are currently presented.

Laser illumination is considered in the table of Fig. 2. The authors restrict their approach to LEDs and do not provide data measured with lasers. Does this mean that this polarization solution compensates specular reflection of LEDs only? What about the roughness /curves in the tissue and their reflection produced by the relief? An important element of novelty would have been testing this approach with excitation lasers and check how the reflections from the speckles are or not reduced with the filter.

Unless the authors can either go deeper into the measurements, their extension to other light sources, provide a huge set of measurements, I think this work lacks novelty and impact to be publishable in this journal of elsewhere. Therefore, I’m afraid I cannot recommend this manuscript to be accepted.

Author Response

Comment 0 :

The authors present measurements showing specular reflection of LEDs as they impact cancer diagnosis due to effects produced on fluorescent/colored imaging systems and how to avoid it with a polarization filter

I feel the approach is quite obvious and I have concerns about the novelty of this work after checking the references 28, 30 and 37. Two of the authors have already published on the subject in this very journal this year; see ref. 30.

The authors focus on study cases and not on accumulating huge date to generate datasets that can lead to rigorous quantification of the specular refection compensation via the polarization.

Answer 0 :

Thank you very much for your interest. We did our best to correct and reflect your comments.

While writing the thesis, I missed the point about the differentiation of [30]. The reviewer gave me a very good point. Therefore, this excellent point will be an opportunity for my thesis to be improved, and I sincerely thank you for such a point.

So, I made up for the lack of detail by recording lines 195-219 (gray) and Figure 6 in detail. Note the following (same as lines 195-219 (gray))

According to the analysis, [30] has similar properties as a method for removing light reflection. The characteristic of [30] is the generation process of diffuse reflection and specular reflection by LED in gastroscope or colonoscope. When the LED is irradiated on the surface of the tissue mucosa, light reflection on the LED occurs due to moisture on the mucosal surface. In this case, light reflection includes diffuse reflection and specular reflection. The characteristic of light reflection by LED is mainly white light, and the wavelength of irradiation and the wavelength of reflection are the same. However, the feature of this study is light reflection by fluorescence expression. When the LED is irradiated to the tissue, a chemical reaction occurs in the tissue by the fluorescent contrast agent. At this time, fluorescence is emitted by a chemical reaction in the tissue. Therefore, diffuse reflection and specular reflection correspond to the fluorescence emission wavelength band, not the LED irradiation wavelength band, as shown in Figure 6 (a). That is, the diffuse reflection and specular reflection generated by the chemical reaction of the fluorescent contrast agent as shown Figure 6 (b) [31]. For this reason, the results of this study are analyzed for light reflection caused by light reflection and fluorescence emission by LED irradiation, and a method to effectively reduce light reflection is studied. Therefore, in the diagnosis process, the visual field of the lesion is blocked by light reflection to reduce the phenomenon that interferes with diagnosis.

Although it may look like a research case, it is a study to eliminate light reflection of chemical reactions that occur in the process of fluorescence expression. The method of previous research is light reflection generated by irradiation of LEDs. Therefore, a method to effectively solve the light reflection caused by fluorescence is the core of this study. In order to solve this essential problem, light reflection characteristics for LED irradiation wavelength and fluorescence expression wavelength were analyzed, and a study was conducted to solve light reflection (refer to lines 195-219 (gray)). Therefore, the results were presented through phantom and animal experiments. The following content is inserted in the body of the paper. please refer to lines of 220 to 227 (sky blue).

[30] LED irradiation reflects the light to the LED irradiation by moisture on the surface of the mucous membrane. This light reflection has a white light wavelength band. However, since the fluorescence expression observes the flow condition of the blood vessel, the light reflection due to the fluorescence expression obscures the field of view of the minute blood flow condition [31]. That is, since light reflection due to fluorescence expression appears in the form of a dot, it blocks the field of view for a minute blood flow condition, which can be very difficult to diagnose. Therefore, we propose a method to solve such fluorescence light reflection.

Thank you very much for your advice and comments so that we can improve the paper. Sorry for the confusion with the picture. So, I modified the figure so as not to cause confusion as shown below and added the content. (Please refer to lines of 110 to 121 (black) and figure 2 (a))

Surgical microscopes are classified into fixed type, pen type, and hand-held type, as shown in Figure 2 (a) [8, 20]. These devices are used for observation imaging during the surgery. These structural features are built in the LED to brighten the dark field of view around the imaging camera, as shown in Figure 2 (b), and the observation method using a fluorescent contrast agent secures the field of view to effectively observe the tissue from the operating microscope, whereas the light from the LED brightly illuminates the tumor [1, 2, 12, 26]. In addition, since the fixed type of probe is large and heavy, it is mounted on the arm and fixed. Therefore, movement in the operating room is not free and the range that can adjust the camera angle is limited. On the other hand, the hand-held and pen type are miniaturized so that the probe can be held by hand. Because of its good mobility and the small size of the probe, it has the advantage of being able to easily access areas that are difficult to image in a fixed device.

Comment 1 :

Laser illumination is considered in the table of Fig. 2. The authors restrict their approach to LEDs and do not provide data measured with lasers. Does this mean that this polarization solution compensates specular reflection of LEDs only? What about the roughness /curves in the tissue and their reflection produced by the relief? An important element of novelty would have been testing this approach with excitation lasers and check how the reflections from the speckles are or not reduced with the filter.

Answer 1 :

Thank you very much for your advice and comments so that we can improve the paper.

We did our best to respond to your comments as follows, and they were reflected in the manuscript lines of 414-433 (pink) and Figure 17.

This study proposes a method to remove light reflection and light reflection generated from LEDs of an endoscope. Of course, when laser is used, light reflection is analyzed to occur steadily [39]. In addition, it is estimated that the optical sources (LEDs and lasers) for all incoherent and coherent will cause light reflection, so the light reflection will result in both diffuse reflection and special reflection, as shown in equations (1)-(6) and Figure 16. This study analyzes light reflection (diffuse reflection and special reflection) by LED irradiation and light reflection (diffuse reflection and special reflection) generated by chemical reaction of fluorescent contrast agents when irradiating LEDs and suggests a method to effectively remove such light reflection. When the LED is irradiated, the light without filter is strongly reflected in the center of the phantom, and the area around the phantom is diffuse reflection, as shown in Figure 16. At this time, if LPF2 (Fig. 14) is rotated (0°-180) in Figure 17, it can be seen that light reflection is suppressed (98.9%) when the angle of rotation (LPF2) is 90° as shown in Figure 17.

In the case of the laser, as illustrated in Figure 17, the light filter generates wide and strong reflections in the center of the phantom. However, if LPF2 (Fig. 14) is rotated (0°-180°), light reflection is suppressed (97%) when the angle of rotation (LPF2) is 90° as shown in Figure 17. The reason is that if the light reflection intensity (IEP) becomes cos290° as shown in Equation (7)-(10), when the rotation angle of F1 is 0° as shown in Figure 10, F2 has an orthogonal relationship with each other, so that the magnetic field (MT) is offset and the electric field (ET) disappears inversely.

Comment 2 :

Unless the authors can either go deeper into the measurements, their extension to other light sources, provide a huge set of measurements, I think this work lacks novelty and impact to be publishable in this journal of elsewhere. Therefore, I’m afraid I cannot recommend this manuscript to be accepted.

Answer 2 :

Added 195-219 (gray) of lines to show novelty and influence. Thank you very much for your generous comments for the development of my paper.

Reviewer 2 Report

This manuscript provides a method to reduce the specular reflection of surgical glare microscope system-generated images, wherein the polarization of the incident light is controlled using the rotation angle of a filter that allows the horizontally polarized light to pass through. The proposed method was evaluated by performing phantom, and small animal experiments. The manuscript is timely and interesting. Although it is well written and organized, there are some minor points that should be considered:

1-     Figure 2 (a) is confusing, it appears like a table. I think it will be better if the contents of the figure are illustrated as a schematic, chart, or block diagram.

2-     Did the authors examine the contrast of the obtained images before and after applying the proposed method, (i.e. using the filter)? If yes, please clarify.

3-     The speckle noise of the obtained images before and after applying the proposed method should be calculated and compared.  

4-     Figure 14 is also presented in a table shape, it will be more appropriate to display it in another presentation style.

5-     Page 15, line 378, please replace “should be suitable” with more appropriate wording. You may say “this method is recommended for the application in …”

6-     Quantified results should be mentioned in the conclusion section.

Author Response

Comment 0 :

This manuscript provides a method to reduce the specular reflection of surgical glare microscope system-generated images, wherein the polarization of the incident light is controlled using the rotation angle of a filter that allows the horizontally polarized light to pass through. The proposed method was evaluated by performing phantom, and small animal experiments. The manuscript is timely and interesting. Although it is well written and organized, there are some minor points that should be considered:

Answer 0 :

Thank you for your advice and advice for your interest and support. We have done our best to respond to your comments.

Comment 1 :

1- Figure 2 (a) is confusing, it appears like a table. I think it will be better if the contents of the figure are illustrated as a schematic, chart, or block diagram.

Answer 1 :

Thanks for the good comments. We strongly agree with your groundbreaking opinion and have revised the contents as follows.

Comment 2 :

Did the authors examine the contrast of the obtained images before and after applying the proposed method, (i.e. using the filter)? If yes, please clarify.

Answer 2 :

Yes. Figure 15 was mirrored to investigate image contrast. (lines of 396 to 403 (green) and Figure 15).

Figure 15 shows the experimental results obtained using the ICG phantom. Here, the yellow marker indicates specular reflection which was observed in images obtained before using the filter. When the rotation angle (θ) of the filter connected to the LED rotated the filter in units of 0° to 360°, a specular reflection of reduced intensity can be observed. Also, when the rotation angle (θ) is 90 and 270°, the specular reflection is sufficiently reduced, and its intensity is analyzed differently depending on the rotation angle (θ) for other results. To obtain reliable results, this imaging testing was performed three times (without LPF and with two different LPF), and were found to produce consistent results. The calculation was performed with the following contents and the speckle noise expression was reflected. Please refer to 447 to 449 (blue), 450 to 478, Eq. (13-17), Figure 18, and Figure 19 such as follows :

It is recommended as a method for fluorescence diagnosis in the future and is expected to be highly utilized. We also believe that these research methods will be fully applicable to the healthcare sector through product design and clinical trials in the future

Comment 3 :

The speckle noise of the obtained images before and after applying the proposed method should be calculated and compared.

Answer 3 :

In order to numerically observe the removal state of light reflection in the resulting images of Figure 15 and Figure 18, it is necessary to obtain the speckle noise contrast evaluation. Result of the phantom as shown in Figure 19. At this time, the beam homogeneity should be observed. In the result, the red waveform is the speckle noise condition without using the LPF, and the blue waveform is the result of the light reflection removal condition using the LPF.

In the results, the phantom on the right is the phantom before and after using the LPF as shown in Figure 18, and the left side is the result of speckle noise conditioning for the phantom before and after using the LPF.

At this time, light reflection occurs before using the LPF, but it can be confirmed that the light reflection is removed when the LPF is used.

Therefore, the probability density (pEP (IƟ=0°)) of the speckle noise before and after using the LPF is the same as the equations (13 and 14). At this time, the parameter (τ) is 0.21 as in equation (15) [40].

Here, pEF (IƟ=0°) without LPF is 0.0057 and pEF (IƟ=90°) with LPF is 0.0000038. The intensity of light reflection (IEP) for the results obtained from these images (see. Fig. 15 or 18) is 100 for the imaging frequency presented in equations (16) and (17).

Comment 4 :

Figure 14 is also presented in a table shape, it will be more appropriate to display it in another presentation style.

Answer 4 :

Figure 14 has been modified to Figure 18.

Comment 5 :

For page 15 (line 378), lines of 447-449 (blue) and 493-501 (blue) are properly expressed. Thanks for pointing it out.

Answer 5 :

For page 15 (line 378), lines of 447-449 (blue) and 493-501 (blue) are properly expressed. Thanks for pointing it out.

Reviewer 3 Report

The experimental results and conclusions are not well demonstrated, specifically:

1) It is suggested that the author fully understand the regularity of the polarization state of reflected light in Fresnel's law, and then improve the scientific interpretation of the experimental results in this paper.  

2) In the context corresponding to fig. 12, there is no any description about LPF3, it is confusing.

3) The fig. 13 and 14 are poor, and the description is also puzzling, so that the phantom experiment seems no so necessary. It's better to give more application results of eliminating specular reflection in animal experiments.

Author Response

Comment 0 :

The experimental results and conclusions are not well demonstrated, specifically:

1) It is suggested that the author fully understand the regularity of the polarization state of reflected light in Fresnel's law, and then improve the scientific interpretation of the experimental results in this paper.

Answer 0 :

Thank you very much for your advice and comments so that we can improve the paper. We did our best to respond to your comments as follows, and they were reflected in the manuscript lines of 146-187 (wine).

Comment 1 :

2) In the context corresponding to fig. 12, there is no any description about LPF3, it is confusing

Answer 1 :

Thank you very much for your comments. I think I missed something during the paper writing process. Therefore, the pointed out points have been supplemented as follows. See Manuscript's lines of 406-411 (red).

Comment 2 :

The fig. 13 and 14 are poor, and the description is also puzzling, so that the phantom experiment seems no so necessary. It's better to give more application results of eliminating specular reflection in animal experiments.

Answer 2 :

I sincerely thank you for your generous compliment. In addition, Thank for your good comments. I positively agree with your groundbreaking comment and Here's the answer: See Manuscript's lines of 493-505 (yellow).

In the experiment process, the imaging test is separated the two section which is divided without LPF and with LPF. The without LPF parts are generation of light reflection imaging (diffuser and specular), and the imaging results is composed of vein with colon, liver (portal and hepatic), derma, and tail, respectively. We observed the blood vessel flow with florescence emission using ICG substance in the vein. Then, the imaging results are generated the light reflection condition (see. Fig. 15 of yellow marker). Thus, the imaging results are necessary to removal of light reflection. Figure 15 (see. under the part) shows the removed the light reflection in the imaging results, To remove the imaging of light reflection, the handmade of a pen type probe is used for LPF which is observed the light reflection removal condition (diffuse and specular), and the observation of organ is the vein with colon, liver (portal and hepatic), derma, and tail, respectively, We can also be enough for suppression of the light reflection. In addition, the rotation angle of 90 degree and 270 degree are excellent removed light reflection.

Round 2

Reviewer 1 Report

The authors have made a clear effort to improve the manuscript and highlight the novelty of this approach.

I think this contribution can be accepted for publication

Reviewer 3 Report

The authors have made improvement and I have no new problems for the time being